# Effects of Biochar-Coated Nitrogen Fertilizer on the Yield and Quality of Bok Choy and on Soil Nutrients

Haiwen Bi [1], Jiafeng Xu [2], Kaixuan Li [1], Kaiang Li [1], Huanling Cao [1,*] and Chao Zhao [1,*]

[1] National Engineering Research Center for Wood-Based Resource Utilization, College of Optical, Mechanical and Electrical Engineering, Zhejiang A&F University, Hangzhou 311300, China; bihaiwenccc@163.com (H.B.); 17858623073@163.com (K.L.); lka0510@stu.zafu.edu.cn (K.L.)

[2] School of Information Sciences and Technology, Zhejiang Shuren University, Hangzhou 311300, China; jiafengxu1996@163.com

[*] Correspondence: 20050115@zafu.edu.cn (H.C.); zhaochao@zafu.edu.cn (C.Z.)

**Abstract:** This study was aimed at problems associated with low fertilizer using efficiency, excessive nitrate content of vegetables, and soil degradation in greenhouse vegetable production. A pot experiment was conducted to assess the effects of applying biochar-coated nitrogen fertilizer (BCNF) on the yield, quality, and nitrate content of bok choy (*Brassica rapa* subsp. *Chinensis*) as well as on soil nutrients in greenhouses. Four treatments were set up as follows: no nitrogen fertilizer application (BA), chemical nitrogen fertilizer application (CK), biochar-coated nitrogen fertilizer application (BCNF, the amount of nitrogen was equal to that of chemical fertilizer), and reduced biochar-coated nitrogen fertilizer application (D-BCNF, the amount of fertilizer was 80% of BCNF). Compared with the other treatments, BCNF treatment increased chlorophyll content, plant height, maximum leaf length, maximum leaf width, and other biological characters of bok choy. Compared with CK treatment, BCNF treatment increased the fresh weight of bok choy by 14.02%, while reducing the root–shoot ratio and nitrate content by 19.1% and 46%, respectively. It was further found that the application of BCNF could effectively increase the content of soil organic matter; reduce the leaching loss of nitrate nitrogen, exchangeable calcium and magnesium; and effectively improve nitrogen use efficiency. Therefore, the application of BCNF can not only reduce the loss of fertilizer nutrients, promote plant growth, and improve fertilizer utilization, but it can also improve soil nutrients, fix carbon, and reduce emissions. It is a new type of environmental protection fertilizer with application prospects.

**Keywords:** biochar-coated nitrogen fertilizer; bok choy; nitrate; soil

## 1. Introduction

Bok choy (*Brassica rapa* subsp. *Chinensis*) has a short growth period, wide adaptability, high yield, and can be produced annually. It is one of the most important vegetables in the world and plays an important role in stabilizing the supply of vegetables. In facility vegetable production, vegetable farmers apply large amounts of chemical fertilizer to improve the yield of vegetables. However, fertilizer use efficiency is only about 30~40%, which not only causes waste of resources but also causes the hardening of soil in facilities and excessive nitrate content in vegetables [1]. The study found that 70~80% of nitrate intake by humans comes from vegetables, and excessive nitrate intake can induce canceration of the digestive system [2]. Food safety is first, and the safe production of grain and vegetables is of great significance [3]. Therefore, the research on developing new green fertilizers, improving fertilizer use efficiency, and reducing nitrate accumulation in vegetables has attracted more and more attention [4]. In this context, biochar-based fertilizer, which has both slow-release properties and soil improvement efficacy, has received extensive studied.

Biochar-based fertilizer is a new type of ecological and environment-friendly slow-release fertilizer, which is made of biochar and organic/inorganic fertilizers [5]. It is

also used in agricultural production as a soil conditioner [6]. Biochar-based fertilizer could not only improve the soil and promote crop growth but could also reduce fertilizer nutrient loss and reduce the pollution from chemical fertilizers to the environment [7]. Meanwhile, biochar can effectively reduce soil nitrate nitrogen leaching [8]. The slow-release performance of biochar-based fertilizer depends largely on the combination mode of phosphorus, nitrogen, potassium elements and biochar [9]. The main combination methods include electrostatic adsorption, complexation, mineralization, etc. The slow-release performance of biochar-based fertilizer is affected by the type of biochar, the preparation method of biochar-based fertilizer, and the ratio of biochar to fertilizer, etc. [10].

In recent years, there have been a lot of studies on the application of biochar as a slow-release fertilizer carrier [11]. Due to the high surface area and porosity and a variety of functional groups, biochar exhibits excellent substrate for the preparation of slow-release and control-release fertilizers [12,13]. A new coated biochar-based slow-release fertilizer (CSRF) was prepared by integrated co-pyrolysis and coating technology. The results showed that the coating technology could significantly improve the slow-release performance of phosphorus [14]. Within 30 days, the cumulative release concentration of phosphorus from CSRFs was half of that of un-coated fertilizer [15]. A N–P slow-release fertilizer (PLB-N) was generated by poultry litter (PL) enriching it with phosphoric acid ($H_3PO_4$) and urea. The study showed that PLB-N has potential as a loading matrix to preserve nitrogen availability and nitrogen use efficiency by plants [16]. Compared with uncoated urea, urea coated with biochar significantly reduced ammonia volatilization by 14% and increased nitrogen use efficiency [17]. A study by Jia et al. suggested that biochar-coated urea (BCU) reduces nitrogen loss mainly by reducing $NO_3^-$-N leaching [18].

Extensive studies have shown that the application of biochar-based fertilizer can increase agricultural output. A pot experiment was carried out to study the effects of $MgCl_2$-modified biochar-based slow-release fertilizer (MBSRF) on maize growth. Compared with the results of chemical fertilizer, the plant height, overground dry matter accumulation, underground dry matter accumulation, chlorophyll content, and leaf area of maize under the application of MBSRF increased by 20.1%, 24.2%, 29.9%, 9.43%, and 24.8%, respectively [19]. Dong et al. applied BCNF to paddy fields and found that the application of BCNF reduced nitrogen leaching and runoff loss at seedling and tillering stages of rice and provided more nutrients at both heading and maturity stages of rice [20]. The combined application of straw biochar with nitrogen fertilizer could increase the rice yield and nitrogen use efficiency, as well as increasing the total biomass accumulation of rice [21]. The 6-year field study showed that the application of biochar-based fertilizer could improve soil structure and reduce chemical fertilizer use. Partial replacement of nitrogen, phosphorus, and potassium fertilizer with biochar is an effective measure to improve soil quality, increase rice growth and yield, and reduce fertilizer input in rice production [22]. The interaction between biochar and nitrogen promotes an improvement in the nitrogen use efficiency of plants [23]. Compared with conventional chemical fertilizer, the combined application of biochar and urea reduced the soil nitrogen loss from 52.00% to 25.94%, and the fresh and dry matter accumulation of maize increased by 292% and 283%, respectively [24]. Bai et al. found that compared to unfertilized and conventional fertilization, the co-application of biochar with inorganic and organic fertilizers increased crop yield by 179.6% and 25.6%, respectively [25].

In recent years, some progress has been made in the research on the slow-release mechanism of biochar-based fertilizer. The nitrogen carrying and storage of biochar-based controlled release nitrogen fertilizer (BCRNF) in the internal pores of biochar were identified by scanning electron microscopy and gas adsorption, and the interactions between biochar, urea, and bentonite help the diffusion and infiltration of water, resulting in nutrient retention [26]. BCRNF can improve the release time and rate of nitrogen in water and soil by combining biochar adsorption with polylactic acid (PLA) coating. PLA coating not only significantly affects the N release of BCRNF but also significantly affects the morphology and thermal properties of BCRNF. The higher the PLA concentration, the

lower the nitrogen release and the longer the release time [1]. When nitrogenous fertilizer is used in combination with biochar, nitrogen is retained over a wide range of specific surfaces of biochar. This not only improves nitrogen use efficiency but also reduces nitrogen loss through leaching [27]. The use of biochar-based fertilizer is of great significance for biomass utilization, carbon sequestration, environmental improvement, ensuring food and vegetable security and promoting sustainable agricultural development [28,29].

In summary, significant progress has been made in the research on biochar-based fertilizers in terms of preparation, slow release of nutrients, increase in crop yield, and slow-release mechanisms. However, there are few reports on the application of biochar-based fertilizer in facility vegetables. It is hypothesized that the application of BCNF could improve nitrogen use efficiency, increase the yield and reduce the nitrate content in bok choy in greenhouse vegetable production. Meanwhile, the application of BCNF is expected to improve the soil quality in greenhouses. To test this hypothesis, a pot experiment was employed to compare the effects of applying BCNF on the yield, quality, and nitrate content of bok choy in greenhouses. At the same time, the effects of BCNF on soil nutrients were studied. Comprehensively analyzing the slow-release performance of BCNF and taking effective measures to mitigate the degradation of facility soil can provide theoretical guidance for optimizing the slow-release performance of BCNF and soil improvement, and it is also of great significance to further improve the quality and safety of vegetables and promote sustainable agricultural development. It was expected that the yield of vegetables could be increased, while that of nitrate content could be reduced so as to increase yield, improve quality, and provide consumers with healthier vegetables.

## 2. Materials and Methods

### 2.1. Materials

The experiment was conducted at Guantang Farm in Lin'an, Hangzhou city (26°29′ N, 106°39′ E. Zhejiang Province, China) from June to July 2021. The test soil was taken from the 0~20 cm topsoil of Danmuqiao Village, Xiaoshan, Hangzhou city (26°29′ N, 106°39′ E. Zhejiang Province, China). The basic properties of the soil were pH 6.97, organic matter 1.877%, total nitrogen content 0.087%, ammonium nitrogen ($NH_4^+$-N) 22.4 mg·kg$^{-1}$, nitrate nitrogen ($NO_3^-$-N) 13.60 mg·kg$^{-1}$, available P 26.20 mg·kg$^{-1}$, and available K 188.7 mg·kg$^{-1}$. The soil was dried naturally in the laboratory to remove weeds, stones, and other sundries. Then, the soil was passed through a 2 mm sieve and stored for later use.

### 2.2. Preparation of BCNF

To prepare the BCNF, urea was used as the fertilizer core, biochar from rice husk was used as the coating material, and nano-SiO$_2$-starch-polyvinyl alcohol was used as the binder. The binder preparation was detailed according to a previous study [30]. Biochar was slowly pyrolyzed at 600 °C using rice husk as the raw material, in which the ash content was 66.57%, moisture content was 4.38%, volatile content was 5.69%, and fixed carbon was 23.36%. After cooling, it was passed through an 80-mesh sieve. Biochar accounts for 30% of the total amount of coated fertilizers [31]. Then, the granulated urea was added (particle size 3~4 mm) into the coating machine (BY-300, Changsha Zhuocheng Medical Instruments Co., Ltd., Changsha, China), the heating function was started, the speed was set at 30 r/min. When the temperature of the fertilizer surface had risen to 65 °C, the nano-SiO$_2$-starch-polyvinyl alcohol binder was sprayed onto the surface of urea with a high-pressure spray gun (PQ-2, Baishi Shun Pneumatic Tools Co., Ltd., Hangzhou, China). At the same time, the biochar was added into the coating machine. After, the coating machine was kept rotating for about 5 min to cover the urea surface with biochar, repeating the process until the expected biochar was completely covered.

### 2.3. Experiment Design

Four treatments were set up in the experiment: no nitrogen fertilizer application (BA), chemical nitrogen fertilizer application (CK), biochar-coated nitrogen fertilizer applica-

tion (BCNF, the amount of nitrogen was equal to that of chemical fertilizer), and reduced biochar-coated nitrogen fertilizer application (D-BCNF, the amount of fertilizer was 80% of BCNF). Each treatment had three replicates, and there were 12 treatments in total. A plastic pot with a height of 12.0 cm, an upper diameter of 16.0 cm, and a lower diameter of 12.0 cm was used for pot experiment. In total, 1.5 kg of air-dried soil was put into each pot. Urea, superphosphate, and potassium sulfate were applied according to $N:P_2O_5:K_2O = 0.2:0.15:0.2$ g·kg$^{-1}$ of dry soil. The specific fertilizer application amount is shown in Table 1.

**Table 1.** Fertilizer amount for each treatment group.

| Treatment | Urea (g·pot$^{-1}$) | BCNF (g·pot$^{-1}$) | Superphosphate (g·pot$^{-1}$) | K$_2$SO$_4$ (g·pot$^{-1}$) |
|---|---|---|---|---|
| BA | 0 | 0 | 1.875 | 0.600 |
| CK | 0.469 | 0 | 1.875 | 0.600 |
| BCNF | 0 | 0.670 | 1.875 | 0.600 |
| D-BCNF | 0 | 0.536 | 1.875 | 0.600 |

Before sowing, the soil and fertilizer were fully mixed and put into pots without topdressing. Deionized water was injected to fully moisten the soil and make the humidity reach the maximum field water-holding capacity. Bok choy was sown on 1 June 2021, with 10 seeds in each pot. On 6 June, the three best-growing seedlings were left in each pot. Regular and quantitative watering management was carried out for bok choy. The growth period of bok choy was 40 days, and then it was harvested on 10 July. After harvesting, the biological characters of bok choy were measured; at the same time, the soil was collected to determine its physicochemical properties.

### 2.4. Determination of Biological Characters of Bok Choy

During the growth period of bok choy, the chlorophyll content of the third real leaf of bok choy was measured by a portable chlorophyll meter (SPAD-502 Plus, Konica Minolta, Tokyo, Japan). The plant height of bok choy was measured before harvesting. The root length, the maximum leaf length and width, and the number of leaves were measured at harvesting, and the maximum leaf area was calculated by Equation (1).

$$A = a \times b \times 0.75 \tag{1}$$

where *a* is the maximum leaf length, cm; *b* is the maximum leaf width, cm; and 0.75 is the leaf area coefficient.

After harvesting, the overground and underground fresh weight of bok choy were weighed. Afterwards, the overground and underground parts of bok choy were put into an air-drying oven (DHG-9240A, Shanghai Jinghong Experimental Equipment Co., Ltd., Shanghai, China) at 85 °C to determinate its dry matter accumulation. The nitrate content in bok choy was determined by salicylic acid nitration colorimetry.

### 2.5. Determination of Soil Physicochemical Properties after Bok Choy Harvesting

Soil organic matter, the concentration of $NH_4^+$-N and $NO_3^-$-N, were measured by a soil fertilizer nutrient tachometer (HM-TYD, Shandong Hengmei Electronic Technology Co., Ltd., Weifang, China). Soil exchangeable calcium and magnesium were extracted by $NH_4Oac$ and determined by atomic absorption spectrophotometer (AA-3800, Shanghai Metash Instruments Co., Ltd., Songjiang, China).

### 2.6. Data Processing

The basic data were calculated by WPS **2020**. The significance test was assessed by SPSS **22**. The significant difference was analyzed by $p < 0.05$, and Duncan's test was used

for multiple comparisons. Drawing was carried out using Origin **2022**. The data in the figures/tables are mean $\pm$ standard deviation.

## 3. Results and Discussion

### 3.1. Effect of BCNF on the Growth, Yield, and Quality of Bok Choy

3.1.1. Effect of BCNF on Dynamic Changes of Chlorophyll Content in Bok Choy Leaves

The chlorophyll content of bok choy leaves under different treatments is shown in Figure 1. After 10, 20, 30, and 40 days of planting, the application of nitrogen fertilizer significantly increased the chlorophyll content of bok choy leaves compared with the BA, while the treatments of CK, BCNF, and D-BCNF had no significant difference in the chlorophyll content of bok choy. The treatment of BCNF had the highest chlorophyll content in the leaves among the four treatments during the 20~40th day of the bok choy growth period. Chlorophyll content affects plant photosynthesis, and leaf chlorophyll content is closely related to leaf N nutrition. The results showed that under the same amount of nitrogen fertilizer application, the content of chlorophyll in bok choy under treatment with BCNF was the highest, which was conducive to photosynthesis and plant production. Reducing the amount of BCNF by 20% did not significantly reduce the chlorophyll content in the leaves of bok choy.

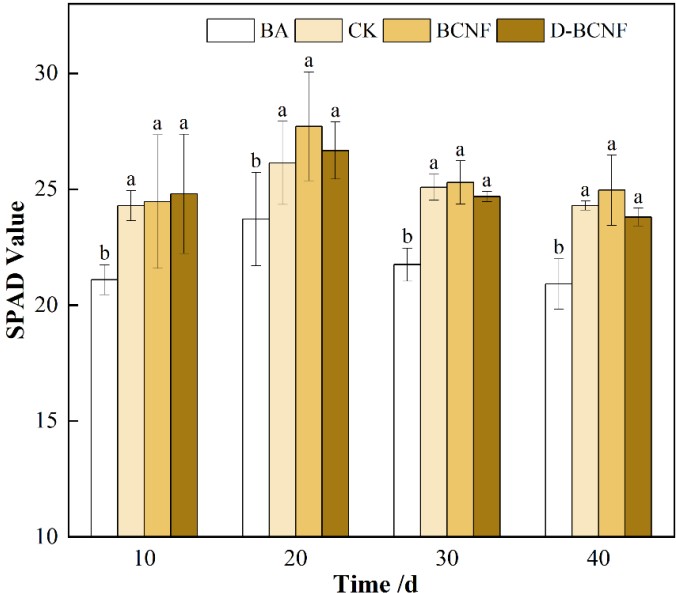

**Figure 1.** Changes in chlorophyll content in the leaves of bok choy under different fertilization treatments. Bars show different letters (a, b) indicating significant differences among treatments ($p < 0.05$). BA: no nitrogen fertilizer application, CK: chemical nitrogen fertilizer application, BCNF: biochar-coated nitrogen fertilizer application, D-BCNF: reduced biochar-coated nitrogen fertilizer application.

Biochar can increase soil moisture and organic matter, while nitrogen fertilizer provides nitrogen for bok choy, which will enhance the synthesis of chlorophyll in bok choy [32]. A study by Xie et al. showed that biochar-based fertilizer could promote the chlorophyll content of fresh maize and promote photosynthesis [33]. A similar research by Khajavi-Shojaei et al. [19] concluded that the application of modified biochar-based slow-release fertilizer (MBSRF) in corn showed a 9.43% higher chlorophyll content in leaves when compared with the application chemical fertilizer. A similar phenomenon was observed in rice leaves with the combined application of straw biochar and controlled-release nitrogen fertilizer [21]. A recent investigation by Chen et al. showed that reducing chemical fertilizer by 20% while combining with biochar did not significantly reduce crop yield and photosynthesis. The results of these studies are similar to results of this study [34].

### 3.1.2. Effect of BCNF on Biological Characters of Bok Choy

Figure 2 shows the plant height of bok choy under different fertilization treatments after harvesting. As can be seen from Figure 2, the plant height of BA was the lowest, which was 10.89 cm, and the heights of those under fertilization treatments were 13.86~14.53 cm. This result shows that the plant height under fertilization treatments was significantly higher than that of BA ($p < 0.05$), which indicates that nitrogen fertilizer could significantly affect the growth of bok choy. The plant height of bok choy under BCNF treatment was the highest, and that of D-BCNF treatment was the same as that of CK treatment. The results show that the growth of bok choy under D-BCNF treatment (reducing the amount of BCNF by 20%) was similar to that under chemical nitrogen fertilizer, indicating that BCNF could effectively improve nitrogen use efficiency. The results indicate that compared with chemical fertilizer, the coating biochar of BCNF effectively reduced the dissolution of nutrients in the early stage and continued to export nutrients in the late stage, thus improving the nutrients use efficiency. Mikos-Szymańska et al. reported that the plant height of spring wheat using biochar compound fertilizer increased by 4~14% when compared with using chemical fertilizer [35]. Yin et al. also found that biochar-based fertilizer can effectively increase ear length and ear diameter of maize by 5.43% and 2.47%, respectively [5].

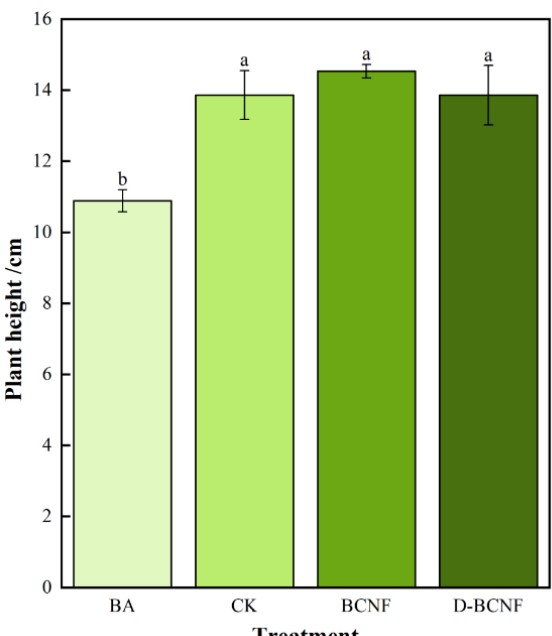

**Figure 2.** Effects of different fertilization treatments on plant height of bok choy. Bars show different letters (a, b) indicating significant differences among treatments ($p < 0.05$). BA: no nitrogen fertilizer application, CK: chemical nitrogen fertilizer application, BCNF: biochar-coated nitrogen fertilizer application, D-BCNF: reduced biochar-coated nitrogen fertilizer application.

Table 2 shows the comparison of leaf number, maximum leaf length, maximum leaf width, and maximum leaf area of bok choy after harvesting under different treatments. It can be seen from Table 2 that the number of leaves was BCNF > D-BCNF > CK > BA, and the maximum leaf length, leaf width, and leaf area were BCNF > CK > D-BCNF > BA. The results show that the number of leaves under fertilizer treatments increased by 7.19~16.47% compared with applying no nitrogen fertilizer (BA), and those of maximum leaf length, maximum leaf width, and maximum leaf area were increased by 20.92~31.16%, 17.85~24.80%, and 42.48~64.32%, respectively. Compared with CK treatment, the maximum leaf length, maximum leaf width, and maximum leaf area of bok choy under D-BCNF treatment were reduced by 3.4%, 0.1%, and 3.6%, respectively, but there was no significant difference between these two treatments ($p > 0.05$), which indicates that a 20% reduction in

BCNF had no significant effect on the growth of bok choy. Khajavi-Shojaei et al. reported that the application of modified biochar-based slow-release fertilizer (MBSRF) in corn increased the corn leaf area by 24.8% when compared with the application of chemical fertilizer [19].

**Table 2.** Effects of different fertilization treatments on the leaf number, leaf length, leaf width, and leaf area of bok choy.

| Treatment | Leaf Number (Piece) | Leaf Length (cm) | Leaf Width (cm) | Leaf Area (cm$^2$) |
|---|---|---|---|---|
| BA | 6.68 ± 0.001 b | 8.89 ± 0.073 b | 6.33 ± 0.112 b | 42.21 ± 12.022 c |
| CK | 7.16 ± 0.532 ab | 11.13 ± 0.023 a | 7.47 ± 0.191 a | 62.36 ± 17.716 ab |
| BCNF | 7.78 ± 0.036 ab | 11.66 ± 0.023 a | 7.90 ± 0.130 a | 69.08 ± 12.924 a |
| D-BCNF | 7.22 ± 0.150 a | 10.75 ± 0.788 a | 7.46 ± 0.424 a | 60.14 ± 25.975 b |

Different letters (a, b, c) after the value indicate significant differences among treatments ($p < 0.05$). BA: no nitrogen fertilizer application, CK: chemical nitrogen fertilizer application, BCNF: biochar-coated nitrogen fertilizer application, D-BCNF: reduced biochar-coated nitrogen fertilizer application.

3.1.3. Effect of BCNF on the Yield of Bok Choy

Figure 3 shows the fresh weight of bok choy under different fertilization treatments after harvesting. It can be seen from Figure 3 that the fresh weight of bok choy under the BCNF treatment was 26.83 g·pot$^{-1}$, while those of under CK and BA treatments were 23.53 g·pot$^{-1}$ and 16.47 g·pot$^{-1}$, respectively. Compared with CK treatment, the fresh weight of bok choy under BCNF treatment increased by 14.02%, while that under D-BCNF treatment decreased by 5.5%. The results show that the application of BCNF could increase the fresh weight of bok choy, thereby increasing the yield. Reducing the amount of BCNF by 20% did not significantly reduce the yield of bok choy. The nitrogen release period of BCNF increased after coating by biochar, which could provide nutrients continuously in the late growth stage of bok choy so as to improve the fertilizer use efficiency. A study reported that the release of nitrate nitrogen and ammonium nitrogen of coated nitrogen fertilizer were 2.5 and 1.5 times slower than those of chemical fertilizers [19]. Compared with urea, biochar-based fertilizer significantly reduced ammonia volatilization by 14% [17]. Mikos-Szymańska et al. proved that the sustained-release property of biochar-coated fertilizer increased. The nutrient leaching rate at 22 days was 65.28%, and nitrogen use efficiency was improved [35].

The overground and underground dry matter accumulation of bok choy after harvesting under different treatments is shown in Table 3. The overground and underground dry matter accumulations of bok choy under BA treatment were 1.21 g·pot$^{-1}$ and 0.093 g·pot$^{-1}$, respectively. The accumulations under CK treatment were 1.54 g·pot$^{-1}$ and 0.065 g·pot$^{-1}$, and those of under BCNF treatment were 1.65 g·pot$^{-1}$ and 0.034 g·pot$^{-1}$. Compared with BA treatment, the overground dry matter accumulation of bok choy harvested from BCNF treatment increased by 35.6%, the underground part decreased by 42.3%, and finally the root–top ratio decreased by 63.3%. Compared with CK treatment, the overground dry matter accumulation of bok choy harvested from BCNF treatment increased by 7.3%, while the underground part decreased by 12.5%. The root-top ratio of bok choy decreased from 0.043 to 0.034. These results indicate that the application of biochar-coated fertilizer could reduce the root-top ratio of bok choy, increase that of overground dry matter accumulation while reducing underground dry matter accumulation, and could thus increase the yield.

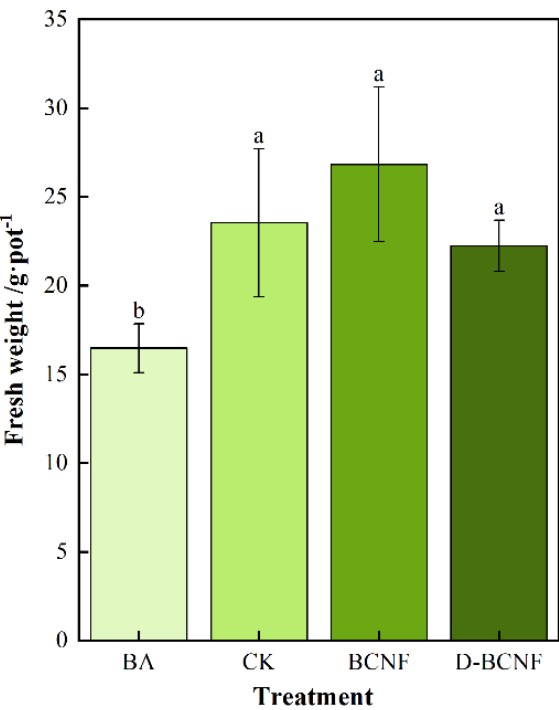

**Figure 3.** Effects of different fertilization treatments on the fresh weight of bok choy. Bars show different letters (a, b) indicating significant differences among treatments ($p < 0.05$). BA: no nitrogen fertilizer application, CK: chemical nitrogen fertilizer application, BCNF: biochar-coated nitrogen fertilizer application, D-BCNF: reduced biochar-coated nitrogen fertilizer application.

**Table 3.** Effects of different fertilization treatments on root–shoot ratio of bok choy.

| Treatment | Dry Matter on the Ground (g·pot$^{-1}$) | Root Weight (g·pot$^{-1}$) | Root Shoot Ratio |
|---|---|---|---|
| BA | 1.2169 ± 0.001 b | 0.1133 ± 0.008 a | 0.0937 ± 0.001 a |
| CK | 1.5376 ± 0.028 a | 0.0654 ± 0.001 b | 0.0425 ± 0.001 b |
| BCNF | 1.6506 ± 0.027 a | 0.0572 ± 0.002 b | 0.0344 ± 0.001 b |
| D-BCNF | 1.4597 ± 0.017 ab | 0.0809 ± 0.002 ab | 0.0572 ± 0.001 b |

Different letters (a, b) after the value indicate significant differences among treatments ($p < 0.05$). BA: no nitrogen fertilizer application, CK: chemical nitrogen fertilizer application, BCNF: biochar-coated nitrogen fertilizer application, D-BCNF: reduced biochar-coated nitrogen fertilizer application.

It is believed that the high root biomass of a plant is caused by low-nitrogen stress under lower fertility treatment [36]. Under low-nitrogen stress conditions, if the relative increase in C/N is higher than 40%, stem growth will be inhibited, root dry weight will increase, and it is possible to increase the length of the axial roots [37]. This is consistent with the research results of this study, that in the absence of nitrogen fertilizer, the soil is in a low-fertility state, which leads to root growth and an increase in root dry matter weight for bok choy namely. Xia et al. reported that the application of biochar nitrogen fertilizer increased the fresh and dry biomass of maize by 292% and 283%, respectively, when compared with no fertilizer application [24]. The application of modified biochar-based slow-release fertilizer (MBSRF) increased overground dry matter accumulation of corn by 24.2% when compared with chemical fertilizer application [19]. The yield increase caused by the application of biochar-based fertilizer was also observed in wheat [35] and rice [22]. This may be because biochar has a great specific surface area, pore structure, and high cation exchange capacity (CEC). It can help enhance the soil's ability to intercept nutrients and be suitable for the growth of soil microbial breeding habitats, improve the metabolic activities of microorganisms, and promote the soil nutrient cycle to increase the yield [38].

### 3.1.4. Effect of BCNF on Nitrate Content of Bok Choy

Figure 4 shows the nitrate content of bok choy after harvesting under different treatments. It can be seen from Figure 4 that the highest nitrate content of bok choy was 1395.5 mg·kg$^{-1}$, which was observed under CK treatment, followed by that under BCNF treatment, 751.9 mg·kg$^{-1}$. With the same amount of CK treatment, the nitrate content of bok choy using BCNF decreased by 46% compared with the use of chemical nitrogen fertilizer. It was concluded that the application of BCNF could significantly reduce the nitrate content of bok choy. D-BCNF treatment also significantly reduced the nitrate content of bok choy, which was about 49% lower than that of CK treatment and about 6% lower than that of BCNF treatment. The nitrogen of biochar-coated fertilizer was slowly released in the soil for crops to absorb, reducing the accumulation of nitrogen in the soil in the form of ammonium nitrogen and nitrate nitrogen, thus reducing the nitrate content of plant [39]. An investigation by Fu et al. revealed that the application of wheat straw-derived (WBF) biochar fertilizer to cabbage increased vitamin C by 31.55% and soluble sugar content by 31.12% and decreased the nitrate content by 35.38% compared with normal compound fertilizer [40]. Some experiments have also shown that coated fertilizer can effectively reduce the nitrate content in plants [41,42]. The application of biochar-coated fertilizer was of great significance in reducing the nitrate pollution of vegetables and reducing the use of chemical fertilizers.

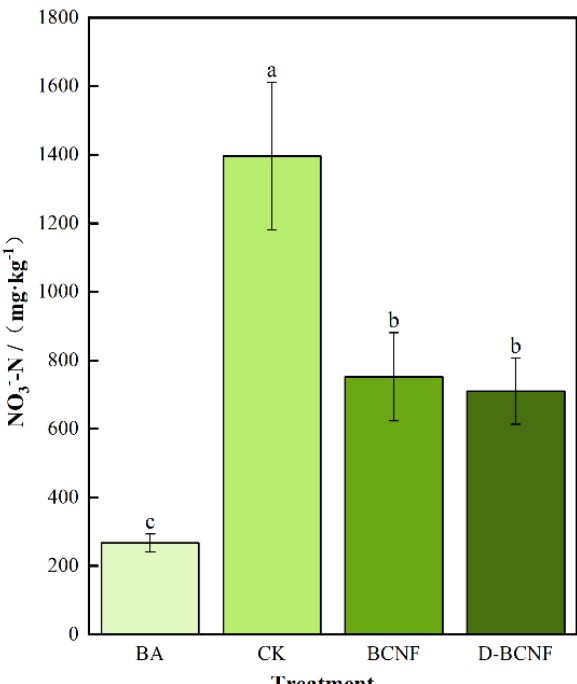

**Figure 4.** Effects of different fertilization treatments on nitrate content in bok choy. Bars show different letters (a, b) indicating significant differences among treatments ($p < 0.05$). BA: no nitrogen fertilizer application, CK: chemical nitrogen fertilizer application, BCNF: biochar-coated nitrogen fertilizer application, D-BCNF: reduced biochar-coated nitrogen fertilizer application.

### *3.2. Effect of BCNF on Soil Nutrients*

### 3.2.1. Effect of BCNF on Soil Organic Matter

Figure 5 shows the soil organic matter content under different fertilization treatments. It can be seen that the soil organic matter content under BCNF treatment was the highest, 15.69 g·kg$^{-1}$, which was 17.05% higher than that under CK treatment. The soil organic matter content under D-BCNF treatment was lower than that under BCNF treatment but still higher than that under CK treatment. There was no significant difference between the treatments of BCNF and D-BCNF. This indicates that BCNF could delay the release

of fertilizer nutrients in the soil, reduce losses such as leaching and fixation of fertilizer nutrients, and improve fertilizer nutrient utilization efficiency. This is consistent with the research by Jiao et al. that biochar-based fertilizer could improve soil organic matter content [43].

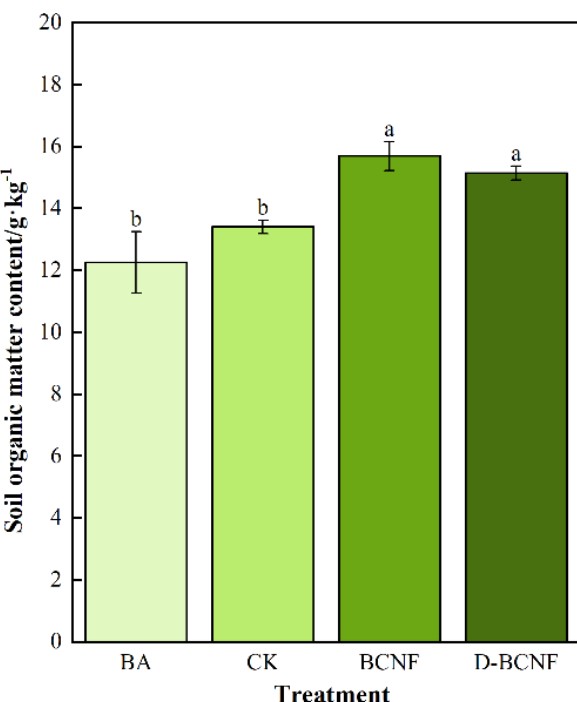

**Figure 5.** Soil organic matter content under different fertilization treatments. Bars show different letters (a, b) indicating significant differences among treatments ($p < 0.05$). BA: no nitrogen fertilizer application, CK: chemical nitrogen fertilizer application, BCNF: biochar-coated nitrogen fertilizer application, D-BCNF: reduced biochar-coated nitrogen fertilizer application.

### 3.2.2. Effect of BCNF on Soil Nitrogen Concentrations

Figure 6 shows the concentrations of $NH_4^+$-N and $NO_3^-$-N in the soil after bok choy harvesting under different fertilization treatments. It can be seen from Figure 6a that the concentration of $NH_4^+$-N under BCNF treatment was the highest, 27.92 mg·kg$^{-1}$, which was 9.9% and 1.9% higher than that under BA and CK treatment, respectively. Overall, there was no significant difference in the concentration of $NH_4^+$-N in the soil under different treatments. At the same time, the concentration of $NO_3^-$-N under BCNF treatment was the highest, 130.17 mg·kg$^{-1}$, which was 1289% and 23.6% higher than that under BA and CK treatment, respectively. There was a significant difference in the concentration of $NO_3^-$-N in the soil under different treatments.

$NO_3^-$-N is not only the main component of soil mineral nitrogen but is also the main form of nitrogen uptake by crops. As a coating material, the surface of biochar contains a lot of charges, which have a strong adsorption effect on $NO_3^-$-N [44,45]. The study by Liao et al. showed that biochar-coated fertilizer can effectively increase the accumulation of nitrogen in the soil surface layer and reduce the leaching of nitrogen [46]. This is consistent with the conclusion of this study that the concentration of $NO_3^-$-N in the soil under BCNF treatment was higher than that under CK treatment. Biochar-based fertilizer mainly reduces nitrogen loss by reducing $NO_3^-$-N leaching, which has also been observed in oilseed rape pot experiments [18]. The structural characteristics, porosity, and chemical interaction with other coating components of biochar play a crucial role in the development of slow-release coatings for biochar-coated fertilizers [20]. The nitrogen is retained on the extensive specific surface of biochar, which not only improves nitrogen use efficiency but also reduces nitrogen leaching loss [27].

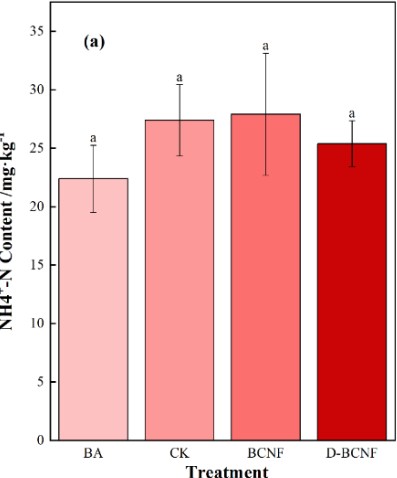
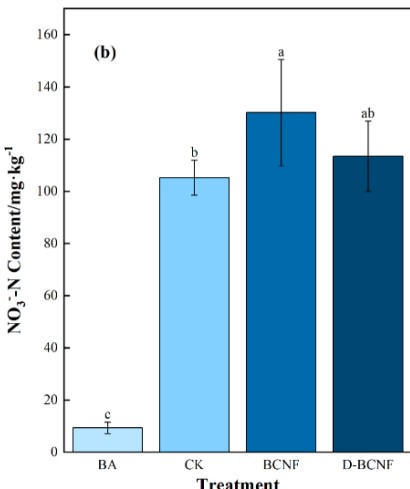

**Figure 6.** Changes of soil ammonium nitrogen (**a**) and nitrate nitrogen (**b**) under different fertilization treatments. Bars show different letters (a, b) indicating significant differences among treatments ($p < 0.05$). BA: no nitrogen fertilizer application, CK: chemical nitrogen fertilizer application, BCNF: biochar-coated nitrogen fertilizer application, D-BCNF: reduced biochar-coated nitrogen fertilizer application.

### 3.2.3. Effect of BCNF on Exchangeable Calcium and Magnesium in the Soil

The concentrations of exchangeable calcium and magnesium in the soil after bok choy harvesting under different fertilization treatments is shown in Figure 7. The concentrations of exchangeable calcium and magnesium in the soil under BCNF treatment were the highest, 2970.0 and 302.5 mg·kg$^{-1}$, respectively, while those under D-BCNF treatment were 2857.5 and 292.5 mg·kg$^{-1}$, respectively, which had no significant difference compared with BCNF treatment. Compared with those under CK treatment, the concentrations of exchangeable calcium and magnesium in the soil under BCNF treatment increased by 8.20% and 22.97%, respectively. Compared with those under BA treatment, the concentrations of exchangeable calcium and magnesium in the soil under BCNF treatment increased by 21.10% and 37.19%, respectively. The results show that the application of BCNF could increase the content of exchangeable calcium and magnesium in the soil, especially for exchangeable magnesium.

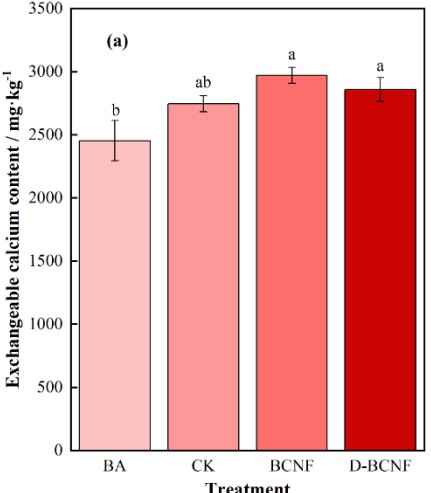
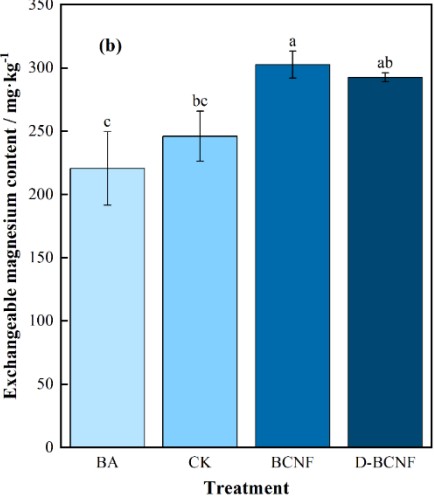

**Figure 7.** Contents of soil exchangeable calcium (**a**) and magnesium (**b**) under different fertilization treatments. Bars show different letters (a, b) indicating significant differences among treatments ($p < 0.05$). BA: no nitrogen fertilizer application, CK: chemical nitrogen fertilizer application, BCNF: biochar-coated nitrogen fertilizer application, D-BCNF: reduced biochar-coated nitrogen fertilizer application.

This may be related to the fact that the addition of biochar could effectively increase the bacterial diversity in the soil and stimulate the activity and growth of microorganisms [47,48]. Exchangeable calcium and magnesium could promote cell growth, chlorophyll synthesis, and photosynthesis as well as improving plant stress resistance [49]. Exchangeable calcium and magnesium are easily leached in the soil profile. Biochar adsorbs exchangeable calcium and magnesium and, thus, reduces their leaching [50]. The concentration increases in exchangeable calcium and magnesium in the soil were conducive to increasing the bok choy yield, which is consistent with the results in Section 3.1.3.

## 4. Conclusions

Compared with the application of the same amount of chemical fertilizer, the application of BCNF increased the fresh weight and overground dry matter weight of bok choy by 14.02% and 7.3%, respectively, while it significantly decreased its nitrate content by 46%. Thus, the application of BCNF in greenhouse vegetable production could increase the yield and improve the quality of vegetables. Reducing the amount of BCNF by 20% did not significantly reduce the yield and quality of bok choy. Meanwhile, the results showed that the application of BCNF could effectively increase the soil organic matter, reduce the leaching loss of $NO_3^-$-N, increase exchangeable calcium and magnesium, and effectively improve nitrogen use efficiency. Therefore, the application of BCNF is of great significance to improve the quality and safety of vegetables and promote sustainable agricultural development.

**Author Contributions:** Writing—original draft, H.B. and J.X.; investigation, H.B.; data analysis, H.B., J.X., K.L. (Kaixuan Li) and K.L. (Kaiang Li); writing—methodology, J.X.; funding acquisition, H.C.; supervision, H.C. and C.Z.; writing—review and editing, H.C. and C.Z. All authors have read and agreed to the published version of the manuscript.

**Funding:** This work was supported by the National Natural Science Foundation of China (32372012) and the Commonweal Project of Science and Technology Agency of Zhejiang Province of China (LGN22C130201).

**Institutional Review Board Statement:** Not applicable.

**Informed Consent Statement:** Not applicable.

**Data Availability Statement:** Data available on request.

**Conflicts of Interest:** The authors declare no conflicts of interest.

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
