# Peer review of "Effects of Biochar-Coated Nitrogen Fertilizer on the Yield and Quality of Bok Choy and on Soil Nutrients"

_sustainability, doi:10.3390/su16041659_

Round 1

Reviewer 1 Report

Comments and Suggestions for Authors

There are many points in the article that must be improved. There was no significant difference in many results of the tests, and it is difficult to convince the reader to draw conclusions on this basis. Soil inorganic nitrogen content during the growing stage of bok choy is required to be measured, and the article lacks this part of the data to determine the characteristics of soil nitrogen changes, as well as soil nitrogen supply, which is essential. Meanwhile, there are many points in the article that must be revised, which are listed below.

Line 30-42: The background of the research is problematic, the problem is not focused enough, what is the problem that the authors wish to solve, whether it is the reduction of nitrate leaching or the reduction of nitrate accumulation in the bok choy or the mitigation of soil degradation problems

Line 43-52: In this paragraph, the effect of biochar-based fertilizer on the destination of soil nitrate should be highlighted

Line 56-59: Why list studies of phosphorus fertilizer?

Line 84-85: Please verify the data, can the increase in yield be almost 3 times compared to chemical fertilizers?

Line 86-87: Please add the control.

Line 103-105: The effects of biochar-based fertilizer on soils have been described in previous parts, so why here again is it said that the effects of charcoal-based fertilizers on soils have been less investigated?

Line 125: Please verify the data, the ammonium nitrogen concentration is too high, the original inorganic nitrogen content is almost equal to the amount of nitrogen applied.

Figure 1 is not clear and all figures are lack of notes.

Line 214: Please mark the figure number

Line 328: This is a confusing result, in the previous results, the reduction of nitrate content in bok choy was due to the fertilizer nitrogen being more in the form of ammonium nitrogen to avoid nitrate from entering the plant, but the soil nitrate content showed that the BCNF treatment had the highest nitrate content, while there was no difference in the ammonium nitrogen content among the treatments, which was confusing.

Line 366: There is a problem with the Figure 7 format.

Comments on the Quality of English Language

The language in the manuscript needs minor revisions.

Reviewer 2 Report

Comments and Suggestions for Authors

68  A pot experiment was conducted carried out to study the effects of MgCl2-68 modified biochar-based slow-release fertilizer (MBSRF) on maize growth.

Usually conducted is not used, but carried out

123  The  basic properties of the soil were: pH 6.97, organic matter 18.77 g/kg, total nitrogen content  0.87   g/kg, ammonium nitrogen (NH4+-N) 128.3 mg/kg, nitrate nitrogen (NO3--N) 13.60  mg/kg, available P 26.20   mg/kg, and available K 188.7 m/kg.(mistake)

Usually the total organic matter and total nitrogen content are presented as % - OM 1.877% and N tot 0.087%.

When we use g/kg or mg/kg we must use International system of units SI – 13.6 g.kg-1, 128.3 mg.kg-1, 26.2 g.kg-1, 188.7 mg.kg-1,

Similar must be the units use in the entire text

Reviewer 3 Report

Comments and Suggestions for Authors

1. Ln14 "Brassica rapa" changes to" Brassica rapa"

2. You have already used BCNF to stand for biochar-coated nitrogen fertilizer, so in the manuscript you should use BCNF in the most of sentences.

3. Do you need to put “,” in front of “respectively” in the most of sentences?

4. Ln151 What do you mean “..a total of 12 treatments..”?

5. Ln169 and Ln185 was or were?

6. In 2.6 Data processing section, please revise this section more detailed.

7. Ln241 change to 17.85-24.80% and 42.48-64.32%,

8. Ln 206-214 why the biochar-based fertilizer could promote the chlorophyll content? Please explain it.

9. Ln341 There was" a" significant ?

Reviewer 4 Report

Comments and Suggestions for Authors

Minor comments on the preparation of the article are included in the attached manuscript
